# Characterization and Anticancer Activity of Biosynthesized Au/Cellulose Nanocomposite from *Chlorella vulgaris*

**DOI:** 10.3390/polym13193340

**Published:** 2021-09-29

**Authors:** Ragaa A. Hamouda, Ahmed I. Abd El Maksoud, Madonna Wageed, Amenah S. Alotaibi, Dalia Elebeedy, Hany Khalil, Amr Hassan, Asmaa Abdella

**Affiliations:** 1Department of Biology, College of Sciences and Arts Khulais, University of Jeddah, Jedda 21959, Saudi Arabia; ragaa.hamouda@gebri.usc.edu.eg; 2Department of Microbial Biotechnology, Genetic Engineering and Biotechnology Research Institute, University of Sadat City, Sadat City 32897, Egypt; 3Industrial Biotechnology Department, Genetic Engineering and Biotechnology Research Institute, University of Sadat City, Sadat City 32897, Egypt; ahmed.ibrahim@gebri.usc.edu.eg; 4College of Biotechnology, Misr University for Science and Technology (MUST), 6th of October City 23546, Egypt; madonna.wageed27@gmail.com (M.W.); daliaebeedy@hotmail.com (D.E.); 5Genomic & Biotechnology Unit, Department of Biology, Tabuk University, Tabuk 71491, Saudi Arabia; a_alotaibi@ut.edu.sa; 6Molecular Biology Department, Genetic Engineering and Biotechnology Research Institute, University of Sadat City, Sadat City 32897, Egypt; hany.khalil@gebri.usc.edu.eg; 7Bioinformatics Department, Genetic Engineering and Biotechnology Research Institute, University of Sadat City, Sadat City 32897, Egypt; amrhassan.nanotechnology@gmail.com

**Keywords:** Au/cellulose nanocomposite, green synthesis, *Chlorella vulgaris*, anticancer, MAPK

## Abstract

Therapeutic selectivity is a critical issue in cancer therapy. As a result of its adjustable physicochemical characteristics, the Au/cellulose nanocomposite currently holds a lot of potential for solving this challenge. This work was designed to prepare a Au/cellulose nanocomposite with enhanced anticancer activity through the regulation of the mitogen-activated protein kinases (MAPK) signaling pathway. Nanocellulose, nanogold (AuNPs), and a Au/cellulose nanocomposite were biosynthesized from microgreen alga *Chlorella vulgaris*. Using UV-Vis absorption spectroscopy, transmission electron microscope (TEM), zeta potential analyzer, and Fourier transform infrared spectroscopy (FTIR), the synthesized nanoparticles were confirmed and characterized. In human alveolar basal epithelial cells (A549 cells), the selectivity and anticancer activity of the produced nanoparticles were evaluated. The cytotoxicity results revealed that the inhibitory concentration (IC50) of the Au/cellulose nanocomposite against A549 cancer lung cells was 4.67 ± 0.17 µg/µL compared to 182.75 ± 6.45 µg/µL in the case of HEL299 normal lung fibroblasts. It was found that treatment with nanocellulose and the Au/cellulose nanocomposite significantly increased (*p* < 0.05) the relative expression of tumor suppressor 53 (p53) in comparison to control cells. They also significantly (*p* < 0.05) decreased the relative expression of the Raf-1 gene. These findings indicate that nanocellulose and the Au/cellulose nanocomposite regulate cell cycles mostly via the motivation of p53 gene expression and reduction of Raf-1 gene expression.

## 1. Introduction

Nanocellulose is made up of a bundle of 1,4-glucan chains with a radial diameter up to 4 nm and has at least one dimension > 100 nm. Nanocellulose was classified into two main types: namely, cellulose nanocrystals (CNCs) and cellulose nanofibrils (CNFs) [1]. Acid hydrolysis can be used to make nanocellulose from cellulose. In addition to being renewable, nanocellulose has advantages of stiffness, excessive strength, dimensional stability, chemical inertness, small coefficient of thermal expansion, little density, the capacity to alter its surface chemistry, a high surface to-volume ratio, and superior tensile strength [2]. Gold nanoparticles (AuNPs) have a large surface area, strong catalytic activity, and high surface energy [3]. When AuNPs are exposed to oxygen or very acidic conditions, they are inert and are not easily oxidized [4].

Nanocomposite materials are obtained by the combination of two or more separate building constituents in one material. They have at minimum one dimension within the nanometer limit (1–100 nm) [5]. They have attracted noteworthy scientific attention due to their phenomenal electrical, mechanical, and barrier properties [6]. These unique characteristics are due to their tiny size, huge surface area, and, of course, the interfacial contact between the two phases [5]. A nanocomposite is produced when nanocellulose and gold nanoparticles are mixed. Cellulose has been utilized as a soft matrix to hold inorganic fillers such as metal nanoparticles (Au, Ag, and Cu) in order to create composites that combine the intrinsic functions of the fillers with the biointerfaces provided by cellulose fibers. Cellulose–fiber-reinforced polymer composites have received much attention because of their low density, nonabrasive, combustible, nontoxic, low cost, and biodegradable properties [7]. The polymer functions as a surface capping, reducing, and/or stabilizing agent when the metal nanoparticles are incorporated or encapsulated inside the polymer matrix [8].

Nanoparticles can be generated by various biological, physical, and chemical methods [9]. Biological approaches are preferred because they are environmentally friendly [10], competitive [11], and more economical than other approaches [12]. Green synthesis is characterized by the use of environmentally friendly materials such as bacteria, fungi, and algae [13]. Green algae have been used for the green synthesis of gold nanoparticles such as *Galaxaura elongate* [14], *Rhizoclonium fontinales*, and *Ulva intestinalis* [15]. *Different species of Ulva, Cladophora, Chaetomorpha, Microdyction, Rhizoclonium*, and *Siphonocladales* also have been used for nanocellulose biosynthesis [16].

Despite scientists’ best attempts to develop curative treatment options, cancer remains the world’s second-largest cause of death [8]. Lung cancer is the world’s largest cause of cancer mortality [17]. A549 cells are human alveolar basal epithelial cells that have developed into adenocarcinoma, which recently replaced squamous cell carcinoma as the maximum frequent histological subtype for all genders and races combined [18]. The occurrence of many mutations that inhibit key signaling pathways is often linked to cancer [19]. One such complex interconnected signaling cascade is MAPK, which is often involved in oncogenesis, tumor proliferation, and drug resistance [20]. RAS/RAF/MEK/ERK genes in the extracellular signal-regulated kinase pathway are commonly affected by cancerous mutations in MAPK pathways [21]. 

Chemotherapy has the disadvantage of low specificity and is limited by dose-limiting toxicity [22]. Nanomaterials have been utilized in the development of cancer drugs, and they have demonstrated significant pharmacokinetic and pharmacodynamic advantages in cancer diagnosis and therapy. Nanoparticles can target cancer biomarkers and cancer cells selectively, enabling for more sensitive diagnostics, early identification with less tissue, long-term monitoring of therapy and tumor burden, and the elimination of only cancer cells [23]. Lately, nanoparticles have been applied in anticancer therapy for numerous kinds of cancer such as Hep2 cell lines [22,24], HT-29 cell lines [25], and breast cancer line MCF-7 [26]. Ramalingam et al. [27] stated that gold nanoparticles have anticancer activity against lung cells A549 cells. He stated that AuNPs arrest the cell cycle at the G0/G2 phase in A549 cells. Liu et al. [28] also found that the growth of A549 cells was inhibited after treatment gold nanoparticles. Mariadoss et al. [29] stated that graphene oxide silver nanocomposite had anticancer activity against the human lung carcinoma cell line. 

Up to date and according to our search in the literature, the impact of the biosynthesized Au/cellulose nanocomposite against cancer cells has not been investigated earlier. Au/cellulose nanocomposites have been biosynthesized from green microalgae *Chlorella vulgaris*. The prepared nanocomposites were characterized. Au/cellulose nanocomposite cytotoxicity to A549 cells was investigated. We also measured the expression of p53 and Raf-1 in A549 cells. The mechanism of anticancer activity of the biosynthesized Au/cellulose nanocomposite against human lung epithelial cells (A549) through the MAPK signalling pathway was investigated. 

## 2. Materials and Methods

### 2.1. Alga

*Chlorella vulgaris* was selected from the algal culture collection of phycology lab, faculty of science, Mansoura University. BG11 nutritive media was used as a medium for algal growth. After 15 days of growth in the stationary phase, the alga was harvested by centrifuge, washed three times by double-distilled water (DD water), and dried at 60 °C until constant weight. Algal water extract was obtained by boiling one gram of alga with 100 mL of distilled water for 30 min; after that, it was filtrated.

### 2.2. Extraction of Cellulose from Alga

A mixture of toluene/ethanol (68:32) (*v*/*v*) was added to *Chlorella vulgaris* dry powder (5 g) and remained for 24 h using magnetic stirring. After 24 h, the mixture was filtrated; to eliminate hemicelluloses, the precipitate was treated with 4% NaOH for 2 h at 80 °C. Then, bleaching was achieved with sodium hypochlorite (10%) at pH 4.8 and magnetic stirring for 2 h at 70 °C. The cellulose pellets were flushed several times with water until the pH of the washing water was 7, collected by centrifugation, and dried by lyophilization [30]. 

### 2.3. Synthesis of Nanocellulose

Five percentage cellulose solution was added to 65% of H_2_SO_4_. To obtain the best yield, the duration and temperature were set to 45 °C for 45 min. After that, the suspension was centrifuged at 10,000× *g* rpm for 10 min and the residue was washed by distilled water three times to eliminate extra sulfuric acid. The suspension was dialyzed against distilled water until the pH becomes 7. The nanocellulose was stored at 4 °C until it was needed [31].

### 2.4. Synthesis of AuNPs 

One mM of AuCl_4_ is dissolved in 90 mL DD water; then, ten mL of the algal water extract was added dropwise to the aqueous AuCl_4_ solution with magnetic stirring at 50 °C. The mixture was stirred until it changed color from yellow to purple, indicating the production of gold nanoparticles. A UV-vis spectrophotometer (Shimadzu 2450, Kyoto, Japan) was used to monitor the reduction of Au^+3^ ions in the solution at regular intervals. After the gold nanoparticles were formed, the suspended solution was centrifuged at 10,000× *g* rpm (Beckman Coulter Avanti J-26SXPI, Brea, CA, USA) for 15 min, and the acquired pellets were redisposed in DD water to eliminate any biomass that had not interacted. The methods of redispersion were repeated twice to get an improved separation of nanoparticles [32].

### 2.5. Au/cellulose Nanocomposite Preparation

The prepared nanocellulose was added to the same volume of AuNPs and stirred for 2 h, to obtain the Au/cellulose nanocomposite. 

### 2.6. Characterization of AuNPs, Nanocellulose, and Au/Cellulose Nanocomposite 

#### 2.6.1. UV-Visible Spectroscopy Analysis

Visual observation was used to record the color change in the algal extract. The bioreduction of gold ions in aqueous solution was measured by taking periodic aliquots of 4 mL and monitoring the solution’s UV-Vis spectra using a Shimadzu UV-1601PC Spectrophotometer (Shimadzu, Kyoto, Japan) at 200–900 nm. 

#### 2.6.2. TEM Analysis 

The shape and size of nanocellulose, AuNPs, and the Au/cellulose nanocomposite were distinguished by TEM. The sonicated sample was placed onto a carbon-coated copper grid, dried overnight in a vacuum, and examined using transmission electron microscopy (JEOL, JEM-2100, Jeol, Tokyo, Japan) operating at 200 kV. 

#### 2.6.3. Zeta Potential Measurement

Using a zeta potential analyzer, the zeta potential value, particle size distribution of the nanoparticles, and polydispersity index (PDI) were determined (Malven Zeta size Nano-Zs90, Malvern, United Kingdom). 

#### 2.6.4. FTIR Measurement

The chemical functional groups of both nanocellulose and the Au/cellulose nanocomposite were determined by an FTIR. For FTIR analysis, the samples were freeze dried and mixed with KBr powder to obtain pellets. The FTIR spectra were collected at resolution of 1 cm^−1^ in the 4000–400 cm^−1^ region using (Nicolete IS10, Thermo Fisher scientific, Waltham, MA, USA).

### 2.7. Cell Lines 

Adenocarcinomic human alveolar basal epithelial cells (A549) (VACSERA, Giza, Egypt) and normal lung fibroblasts (HEL299) (VACSERA, Giza, Egypt) were cultured in RPMI media (Invitrogen, Waltham, MA, USA) supplemented with 4 mM L-glutamine, 100 U/mL penicillin/streptomycin, and 10% *v*/*v* bovine serum (Hyclone SH30071.03) (Hyclone Laboratories, Logan, UT, USA). Proliferative cultures were cultured at 37 °C in a humidified 5% CO_2_ incubator [33].

### 2.8. Cell Viability and Cytotoxic Effects

The cytotoxicity of AuNPs, nanocellulose, and an Au/cellulose nanocomposite on lung cancer A549 cells and on normal lung fibroblasts (HEL299) was determined using an LDH kit. Then, 40 µL (0.39, 1.56, 6.3, 25, and 100) µg/mL of AuNPs, nanocellulose, and an Au/cellulose nanocomposite in RPMI media were incubated for 1 h with 40 µL LDH buffer and 20 L µLDH substrate in a 96-well plate. The relative activity of LDH was measured and calculated using the standard curve given (expressed as experimental LDH release at 490 nm/maximum LDH release at 490 nm ± Standard Deviation (SD). Positive controls were cells treated with 50 and 100 µL of Triton X-100. The IC50 (particle concentration inducing 50% cell mortality) values were calculated by regression analysis using Quest Graph™ IC50 Calculator (AAT Bioquest, Inc., Sunnyvale, CA, USA) [33]. The IC50 values were calculated using Graphpad Prism software (San Diego, CA, USA). IC50 was determined with a non-linear model.

The cytotoxic potential of indicated agents was further achieved by investigating cell morphology using an inverted microscope and accounting for the number of survived cells. The number of survived cells upon treatment was manually accounted based on the removing of the old media of treated cells, washing the cells by PBS, trypsinizing the attached cells, and finally accounting the number of cells using a hemocytometer. 

### 2.9. Total RNA Isolation and cDNA Synthesis

Cells were collected in clean, RNase-free tubes from cell culture plates. TriZol was used to separate total RNA from cells (Invitrogen, Waltham, MA, USA). By dissolving the extracted RNA in RNase-free water, the concentration of all samples was adjusted to a final concentration of 100 ng/µL. Then, using a cDNA synthesis kit, 10 µL of each isolated and purified total RNA was used to produce cDNA (Qiagen, Hilden, Germany). According to the manufacturer’s procedure, total RNA was incubated with reverse transcriptase and poly (dT) primers for one hour at 45 °C, which was followed by five minutes at 95 °C. After that, the cDNA was kept at −20 °C until it was needed [34]. 

### 2.10. Quantitative Real-Time Polymerase Chain Reaction Analysis 

The Raf-1 and p53 genes’ expression levels were determined using quantitative real-time polymerase chain reaction (q-RT-PCR). The resulting cDNA was utilized as a template for PCR amplification utilizing DNA polymerase enzyme and one or more gene-specific primers. The QuantiTect SYBR Green PCR Kit was used to detect the relative gene expression of Raf-1 and p53 (Qiagen, Germantown, MD, USA). Oligonucleotides specific for each individual gene, p53-for: 5′-GCGAGCACTGCCCAACAACA-3′, p53-Rev-5′-GGTCACCGTCTTGTTGTCCT-3′, Raf-1-for: 5′-TTTCCTGGATCATGTTCCCCT-3′, Raf-1-Rev: 5′-ACTTTGGTGCTACAGTGCTCA-3′. Levels of GAPDH were amplified using specific oligonucleotides, GAPDH-5′-TGGCATTGTGGAAGGGCTCA-3′ and GAPDH-Rev-5′-TGGATGCAGGGATGATGTTCT-3′, which was used for normalization as an internal control. In the RT-PCR program, the following parameters were used: 95 °C for 4 min, 40 cycles (94 °C for 45 s, 58 °C for 30 s, and 72 °C for 45 s), and 4 °C holds. Ct equations were used to evaluate the indicated Ct values [34].

### 2.11. Flowcytometry Analysis (FCM)

FCM was used to assay Raf-1 and p53 protein levels in treated cells. FCM is a useful method for quickly detecting and identifying cells based on their light scattering and fluorescence properties [35].

In a 6-well plate, A549 cell lines were seeded at a concentration of 200,000 cells per well in 2 mL of RPMI medium and incubated overnight in a CO_2_ incubator. The treated cells were washed in phosphate-buffered saline (PBS), collected in PBS, and centrifuged at room temperature (RT) for 5 min at 5000× *g* rpm. The supernatant was decanted, and the pellet was resuspended in PBS with triton X-100 (permeabilization step) before centrifugation as defined previously. The supernatant was extracted, and the pellets were resuspended in PBS containing 1% Bovine Serum Albumin (BSA) and 1:100 diluted primary antibodies (mouse monoclonal antibody for Raf-1 and rabbit monoclonal antibody for p53) (Invitrogen, Waltham, MA, USA), followed by one hour at room temperature incubation. After centrifugation, the pellets were washed three times in PBS before being incubated in the dark for 30 min at room temperature with secondary antibodies, either goat anti-rabbit or goat anti-mouse (Alexa Fluor 488) at a dilution of 1:100. The cells were centrifuged, the supernatant was removed, and they were washed as mentioned previously. Finally, the pellets were re-suspended in 500 µL PBS. The relative protein expression was investigated using the FCM (Becton Dickinson FACSCalibur Device, Franklin Lakes, NJ, USA) according to the recommended protocol [20]. 

### 2.12. Statistical Analysis

Final graphs and histograms for our data were generated in Microsoft Excel. The significance of all data given by real-time PCR analysis was investigated using the Student’s two-tailed *t*-test. Using the formulae provided by Khalil et al. [36], qRT-PCT data were processed with SDS 2.2.2 software to obtain Ct values for possible gene expression.

## 3. Results and Discussion

### 3.1. Nanoparticle’s Characterization

#### 3.1.1. Visual Inspection and UV Absorbance Spectroscopic Studies

Nano metals display notable spectral characteristics by surface plasmon resonance (SPR) because of the mutual vibration of free electrons with light wave resonance induced by the sizes and shapes of the biosynthesized NPs. Since SPR initiates an extinction spectrum related to the size, shape, and aggregation of AuNP, UV-vis spectroscopy is a very valuable technique that permits assessment of gold nanoparticle’s size, shape, and aggregation level [37].

The aqueous extracts of alga were pale green, and the gold chloride solution was a pale-yellow color. In the beginning, the faint yellow color was obtained after the addition of *Chlorella vulgaris* algal extracts to the gold chloride solution. After half an hour with heating at 50 °C and stirring, the solution was changed and converted to purple, revealing the formation of AuNPs (Figure 1A). The color of gold nanoparticles can differ from red to blue, which is contingent on the size and shape of the Au nanoparticles [38]. The blue color was formed related to the reduction of Au (III) to Au (0) and subsequent improvement formation of Au NPs intra and/or extracellularly [39]. 

The UV-Visible spectra of biosynthesized AuNPs by *Chlorella vulgaris* showed intense peaks with strong SPR at 534 nm (Figure 1B). The broad peak denotes the presence of large-sized nanoparticles. The longitudinal excitation of surface plasmon resonance of NPs caused this band blue shift [40]. Mie theory investigated the spherical shape and small size of NPs produced in the reaction mixture that was recognized by the formation of a single SPR band for AuNPs [41]. Oza et al. [42] reported that the SPR band of AuNPs synthesized by *C. pyrenoidusa* was observed at 530 nm at pH 8 and centered at 540 nm at pH 10.

#### 3.1.2. Transmission Electron Microscopy (TEM)

TEM images of AuNPs denote that those nanoparticles biosynthesized by *Chlorella vulgaris* were a spherical shape, having a smooth surface, and well-dispersed (Figure 2A). The particle size of AuNPs was in a range from 13 to 16 nm. The same results were obtained by Oza et al. [42], who stated that AuNPs biosynthesized by *C. pyrenoidusa* were in the range from 25 to 30 nm. Parial and Patra [39] also stated that the TEM image of AuNPs biosynthesized by *Phormidium tenue* and *Ulva intestinalis* showed that the nanoparticles were irregular in shape and were having a particle size of 14.84 nm. The size of AuNPs synthesized from brown algae (*Sargassum tenerrimum* and *Turbinaria conoides*) varied from 12 to 57 nm with a typical size of about 27.5 nm [32]. AuNPs that were biosynthesized by *Lyngbya majuscule* were spherical, and the diameter size was 41.7 ± 0.2 nm [43].

Cellulose extracted from *Chlorella vulgaris* was converted into cellulose nanofibrils (CNF) and characterized by TEM image, as presented in Figure 2B, C. As presented in the image, CNF was rod-shaped with diameters ranging from 28.43 to 82.01 nm and well-dispersed. Lani et al. [44] stated also that nanocellulose fiber is rod-shaped; some of them aggregated in the form of bundles while the others were well-separated, and the diameter ranged from 4 to 15 nm. Othman et al. [45] reported that the accumulation of nanoparticles is generally related to the Van der Waals attraction forces between nanoparticles.

A TEM image of Au/cellulose nanocomposite denotes that the shape of the Au/cellulose nanocomposite was spherical, having a smooth surface with aggregation. The particle size of AuNPs was in the range of 113.33 to 203.50 nm (Figure 2D).

#### 3.1.3. Particle Size and Zeta Potential Analysis

The size and zeta potential of the synthesized nanomaterials are essential physiochemical features for nanoparticles. From the zeta potential results shown in Figure 3A, it was obvious that the surface charge of the Au/cellulose nanocomposite biosynthesized by green alga *Chlorella vulgaris* is negative. The average ZP was −13.6 mV at pH 6.5 (Figure 3A). The negative values elucidate the repulsion among the nanoparticles and thus the accomplishment of superior constancy of the Au/cellulose nanocomposite, which prevents agglomeration in aqueous solutions [46]. Major positive or negative data of nanocrystals determined by zeta potential accurately designate the physical constancy of nanosuspensions because of the electrostatic repulsion of unique particles [47].

The average particle size of the Au/cellulose nanocomposites was found to be 114 ± 0.43 nm, PDI:0.442 (Figure 3B). PDI ranges from 0.01 for monodispersed particles to 0.5–0.7 for polydispersed particles. Furthermore, values greater than 0.7 imply polydispersed samples with a wide size distribution. The samples below 0.7 are monodisperse, and the samples above 0.7 may be polydisperse [48,49]. The small PDI of the Au/cellulose nanocomposite indicated the uniformity of the sample and that the sample is well monodispersed. Danaei et al. [50] reported that smaller PDI supports a consistent organization.

#### 3.1.4. FTIR Studies

Seven peaks are present in the Au/cellulose nanocomposite: 3453.88, 2082.74, 1639.2, 1432.65, 1272.79, 1197.58, and 613.252 cm^−1^ (Figure 4).

The strong and broad band observed at 3453.88 cm^−1^ indicated the incidence of a primary amine O-H band and polyphenolic O-H group [25]. The hydroxyl groups exist on the polysaccharides and monosaccharides of the algal material, which might be included in the production of gold nanoparticles [51]. They were concerned about the bio-reduction of Au (III) ions into Au (0) [52]. Peaks at 2082.47 cm^−1^ may be related to C-H bending in aromatic compounds according to Nandiyanto et al. [53]. The band at 1639 cm^−1^ corresponds to amide linkages between amino acid residues in protein and the synthesized gold nanoparticles [54]. The amide linkage is most likely involved in the reduction of gold ions to nanoparticles and the conservation of gold nanoparticles in the medium [55]. The band at 1432 cm^−1^ was due to N–O symmetric stretch nitro groups [56]. A C-O stretching carboxylic acid group was allocated at 12,472 cm^−1^. The band at 670 cm^−1^ was due to alkyl halides [40]. The polysaccharide and protein components in the alga extract served as both a reducing and stabilizing agent for the AuNPs [57]. 

### 3.2. Cytotoxic Effect of Nanocellulose and Au/Cellulose Nanocomposite on A594 Cells

Figure 5 demonstrated the morphological changes of cells after treatment with 100 µg/mL (AuNPs, nanocellulose, and Au/cellulose nanocomposite). When compared to the characteristic adherent shiny spindle-shaped control cells, disruption of the cell monolayer, rounding and shrinking of the cells, as well as characteristic changes of cell death such as granulation, blebbing, and nuclear fragmentation were visible in the treated cells [8].

Figure 6 also indicated a significant decrease (*p* < 0.05) in the number of living cells treated with 100 µg/mL (AuNPs, nanocellulose, and Au/cellulose nanocomposite) compared to cells treated with DMSO and untreated cells. The cytotoxic potential of nanocellulose has also been reported by Stoudmann et al. [58]. 

Based on the cell viability rate, for normal lung fibroblast (HEL299) treated with AuNPs, cellulose, and Au/cellulose nanocomposite, IC50% values were 95.668 ± 3.59, 2055.9 ± 51.19, and 182.75 ± 6.45 µg/mL respectively. Meanwhile, in the case of lung cancer cells A549, IC50% were 13.43 ± 0.76, 86.61 ± 3.21, and 4.67 ± 0.17 µg/mL, respectively (Figure 7 and Table 1). 

These studies showed that AuNPs, nanocellulose, and an Au/cellulose nanocomposite are safe in normal cells, indicating that these reagents might be used to create effective anticancer compositions [59].

### 3.3. Effect of Nanocellulose and Au/Cellulose Nanocomposite on Raf-1 Gene and p53 Gene Expression

One of the most often mutated genes in human cancers is the p53 gene, which encodes the phosphorylated P53. According to research, half of all tumors had inactivated phosphorylated P53. The phosphorylated p53 protein controls a variety of biological processes, including cell cycle, apoptosis, angiogenesis, and immune response [60]. In human cancer, the p53 tumor suppressor gene is the most frequently altered gene [61]. The p53 suppressor gene encodes the transcription factor p53, which controls cell cycle initiation, senescence, differentiation, DNA repair, and apoptosis [62]. p53 may regulate the cell cycle by limiting cellular proliferation either by cell cycle arrest or apoptosis [63].

The overexpression or mutation of the Raf-1 gene can lead to cancer growth. Raf-1 mutation has been discovered in a variety of cancers, including thyroid, prostate, and bladder cancer [64]. The Raf-1 protein, which is encoded by the Raf-1 gene, is involved in the MAP kinase/ERK signaling cascade, which controls several essential cell functions [65]. It had been found that overexpression of this gene or its hyperactivity can interfere with the RAS/MAPK pathway, which can result in many serious developmental disorders [66,67], and also it can lead to various cancer types [68].

The steady-state mRNA of p53 and Raf-1 in A549 treated cells was quantified by using q-RT-PCR. It was found that treatment with nanocellulose and Au/cellulose nanocomposites significantly increased the relative expression of p53 (*p* < 0.01) compared to control cells (Figure 8A). Meanwhile, in cells treated with nanocellulose and Au/cellulose nanocomposites, the relative expression of the Raf-1 gene has been significantly decreased (*p* < 0.01) (Figure 8B). These results suggest that activation of the p53 gene and suppression of the Raf-1 gene are the most likely mechanisms by which nanocellulose and the Au/cellulose nanocomposite regulate cell cycles. 

The percentage of positive events was linked to the relative fluorescent intensity of the Alexa Fluor signal to measure the amount of pho-P53 stabilization in cells treated with nanocellulose and Au/nanocellulose composites against DMSO. Regarding nanocellulose and nanocellulose/nanogold composite pre-treatment, expression of the Raf-1 corresponding protein was identified in 5% and 40% of the cell population, respectively. In the case of p53, 80% and 30% of investigated cells showed positive signaling regarding the same treatment (Figure 9).

MAPK signaling is normally activated at the cell surface, where it stimulates a variety of protein kinases before modulating the transcription factors [69]. The MAPK pathway controls many cellular functions, including cell differentiation, proliferation, programmed cell death, and inflammation [70]. The Raf protein kinase, MEK1/2, and the transcription factor ERK1/2 are all included in the MAPK/ERK signaling pathway. The activation of mutant K-Ras and mutant B-Raf proteins activates this pathway in cancer cells [71]. Raf protein kinases are the essential factors of the MAPK pathway that regulate cell division. The stimulation of Raf kinase proteins is needed for the activation of the upstream Ras protein, which in turn activates the other downstream factors MEK1/2 and ERK1/2 [72]. Nanoparticles can block MAPK signaling, which is required for cancer cell death, by focusing on mutant K-Ras and mutant B-Raf proteins, which are responsible for limiting programmed cell death and increasing TGF-b and IL-8 production [73]. 

Anti-apoptotic proteins including Akt and mTOR gene expression were significantly affected by gold nanoparticle therapy. To induce apoptosis in cancer cells, these proteins, as well as MAPK/ERK, must be inhibited [74,75]. Early and late apoptosis are caused by treatments that involve the Pi3K/Akt/mTOR pathway and the MAPK/ERK enzyme [76]. Gold nanoparticles acquiring anticancer properties are known for their potential ability to slow down the activities of abnormally expressed signaling proteins, such as Akt and Ras, cytokine-based therapies, and DNA- or protein-based vaccines against specific tumor markers, and tyrosine kinase inhibitors, which exhibit a consistent antitumor effect [40]. 

Raf inhibitors, on the other hand, have short-term therapeutic effectiveness and, via therapy, enhance enigmatic tolerance [77]. Axitinib, Lenvatinib, and Cabozantinib are examples of angiogenesis inhibitors that have been licensed based on their efficacy in regulating vascular endothelial growth factor (VEGF) and its receptors [77]. Renal dysfunction, stroke, atherosclerosis, and brain syndrome, as well as reversible posterior leukoencephalopathy, are all serious side effects of VEGF inhibitor therapy [78]. As a result, finding a safe and successful cure for cancer cells is critical in combating the disease.

Being polysaccharides, the anticancer activity of nanocellulose was studied using in vitro cell line studies, which have been tested in vivo in appropriate animal models. Cell cycle arrest, depolarization of the mitochondrial membrane, the nitric oxide pathway, and immunomodulation were all found as similar processes [79]. The anticancer action of the nanocomposite can be connected to the induction of apoptosis in MCF-7 cells via blocking the activation and nuclear translocation of TNF-mRNA and protein expression. TNF-induced inflammation and toxicity are both exacerbated by AgNPs-EC [80,81].

## 4. Conclusions

Nanoparticles have led to rapidly increasing applications in different fields. The purpose of this study is to evaluate the possibility of using Au/cellulose nanocomposites as antitumor therapy. Micro-green algae *Chlorella vulgaris* represented an eco-friendly method of Au/cellulose nanocomposites biosynthesis. They were confirmed by numerous characterizations studies such UV-vis, TEM analysis, FTIR, and zeta potential measurement. The in vitro cytotoxicity showed that the biosynthesized Au/cellulose nanocomposites are nontoxic over normal lung fibroblasts (HEL299). They exert significant cytotoxicity to lung cancer cells (A549) in a dose-dependent fashion in the concentration range of 0.39–100 μg/mL. The anticancer mechanism was elucidated on lung cancer cell (A549). They were having regulatory effects on sustained MAPK signaling. We analyzed the mRNA expression levels of p53 and Raf-1 in response to Au/cellulose nanocomposites exposure in A549 cells, because apoptosis is controlled through these pathways. Quantitative real-time PCR results showed that Au/cellulose nanocomposites significantly increased the relative expression of p53 gene, while it significantly decreased that of the Raf-1 gene. Activation of the p53 gene and suppression of the Raf-1 gene are the most likely mechanisms by which they regulate cell cycles. In summary, our results indicate that Au/cellulose nanocomposites possesses anti-proliferative effects against cancer cells. Au/cellulose nanocomposites may provide a relatively non-toxic therapy for cancer. These findings are encouraging for the development of innovative therapeutic methods for the future treatment of cancer using safe materials, which may overcome adverse effects associated with chemotherapy. Using Au/cellulose nanocomposites in experimental animals to assess their effect in an in vivo system or even in different cell lines will require more comprehensive research.

## Figures and Tables

**Figure 1 polymers-13-03340-f001:**
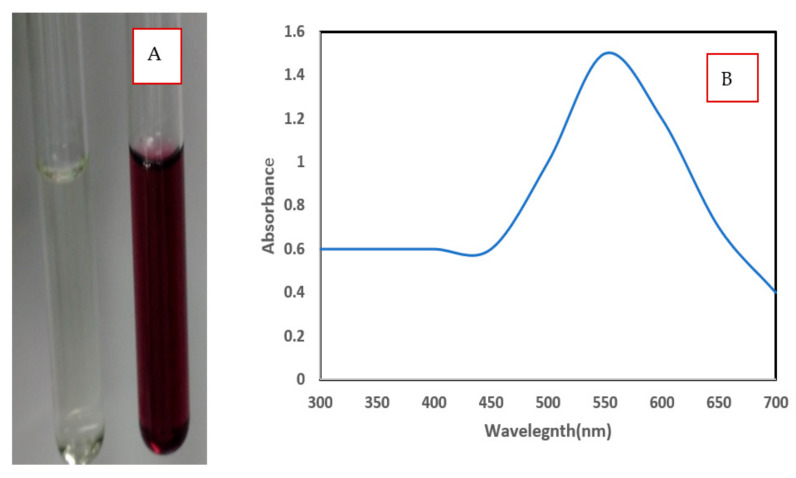
(**A**) The synthesis of AuNPs using green alga *Chlorella vulgaris* was confirmed by changes in solution from colorless to purple color, (**B**) UV-Vis spectra of AuNPs synthesized using green alga *Chlorella vulgaris*.

**Figure 2 polymers-13-03340-f002:**
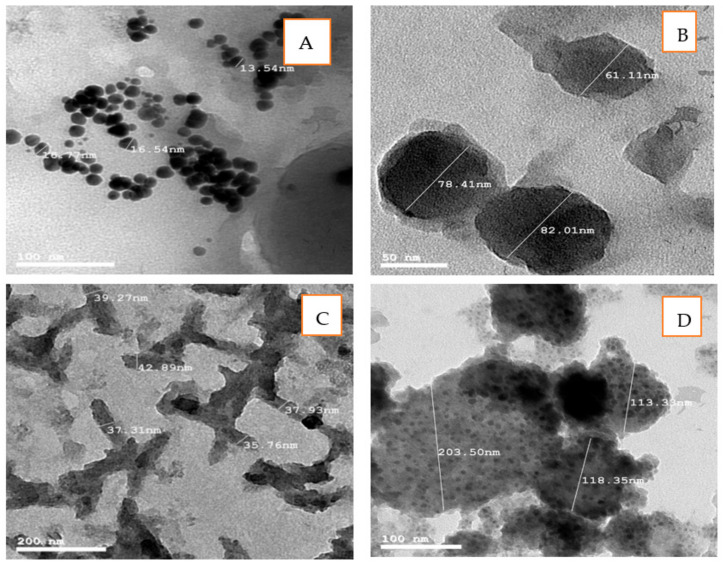
TEM image of (**A**) AuNPs biosynthesized by *Chlorella vulgaris*; (**B**) Horizontal section nanocellulose fibril biosynthesized by *Chlorella vulgaris*; (**C**) nanofibrils distributions; (**D**) Au/cellulose nanocomposite.

**Figure 3 polymers-13-03340-f003:**
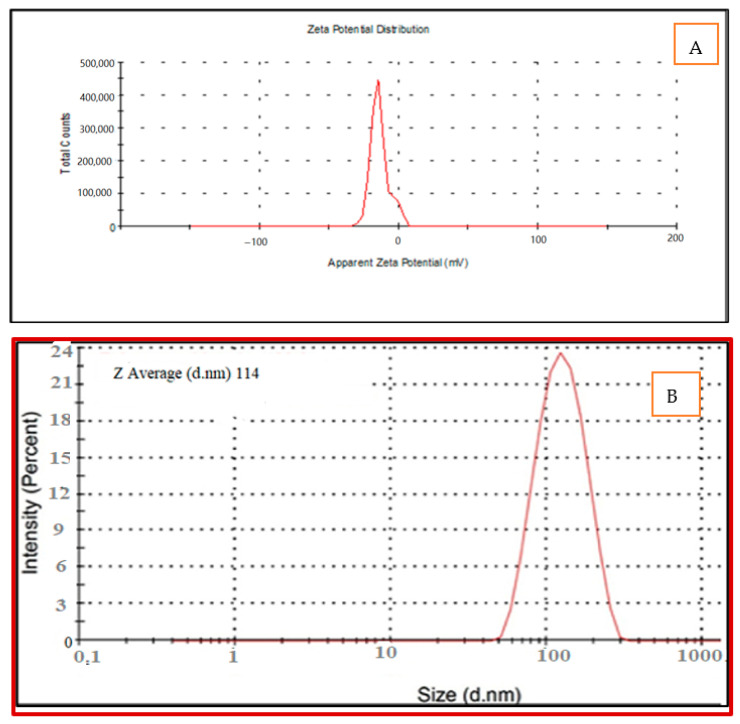
(**A**) Zeta potential analysis (**B**) Particle size distribution of Au-cellulose nanocomposite biosynthesized by *Chlorella vulgaris*.

**Figure 4 polymers-13-03340-f004:**
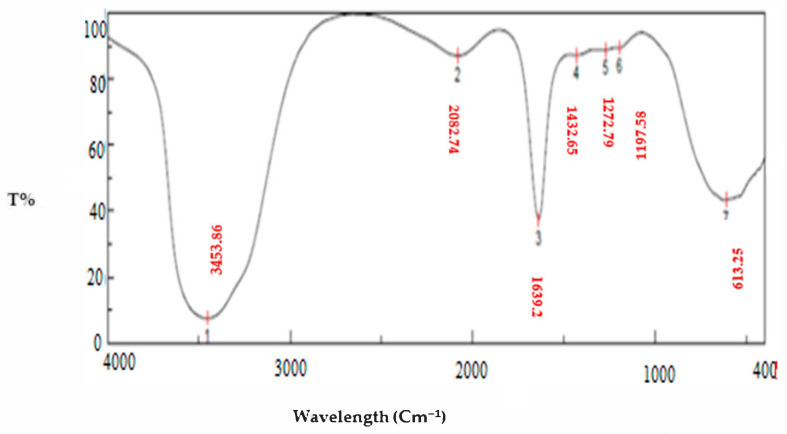
FTIR spectroscopy of the Au-cellulose nanocomposite composite.

**Figure 5 polymers-13-03340-f005:**
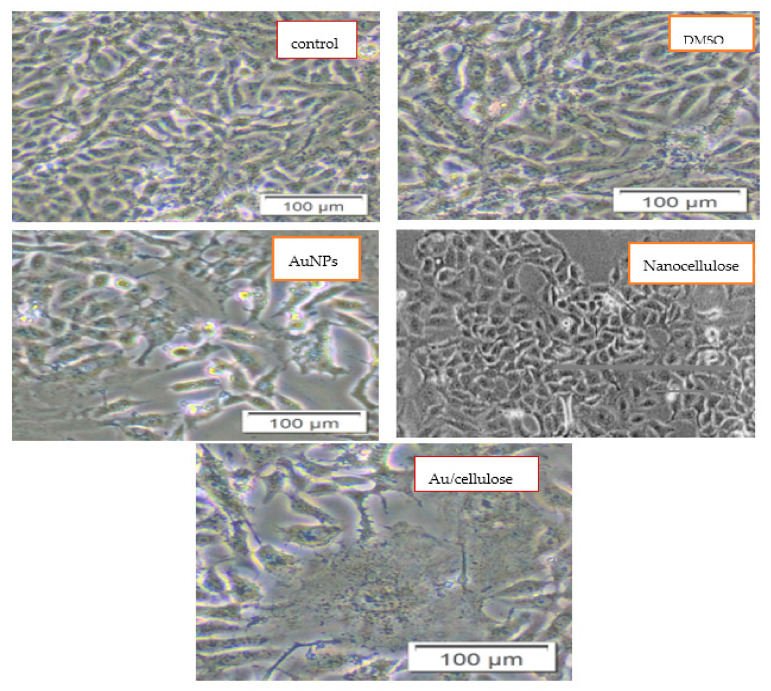
Representative cell images reveal the cell viability of A549 cells that were pre-treated with 100 µg/mL AuNPs, nanocellulose, and Au/cellulose nanocomposite in comparison with DMSO-treated cells and control cells.

**Figure 6 polymers-13-03340-f006:**
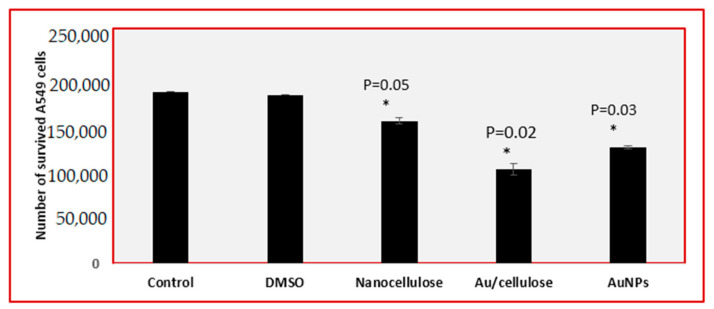
Number of survived A549 cell lines that were pre-treated with 100 µg/mL AuNPs, nanocellulose, and Au/cellulose nanocomposite in comparison with DMSO-treated cells and control cells. * *p* < 0.05 was considered statistically significant.

**Figure 7 polymers-13-03340-f007:**
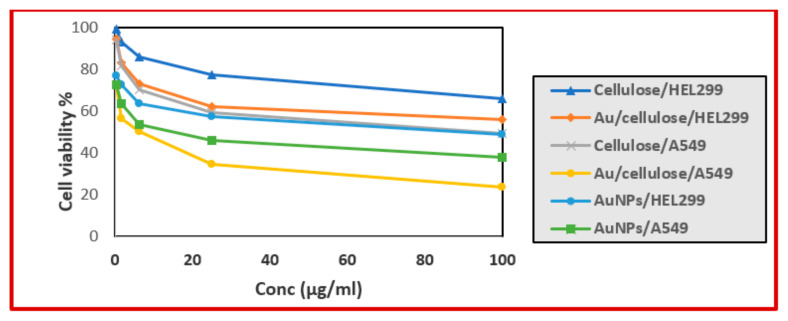
Cell viability rate of normal lung fibroblast HEL299 and lung cancer cells A549 that were pre-treated with different concentrations of AuNPs, nanocellulose, and Au/cellulose nanocomposite.

**Figure 8 polymers-13-03340-f008:**
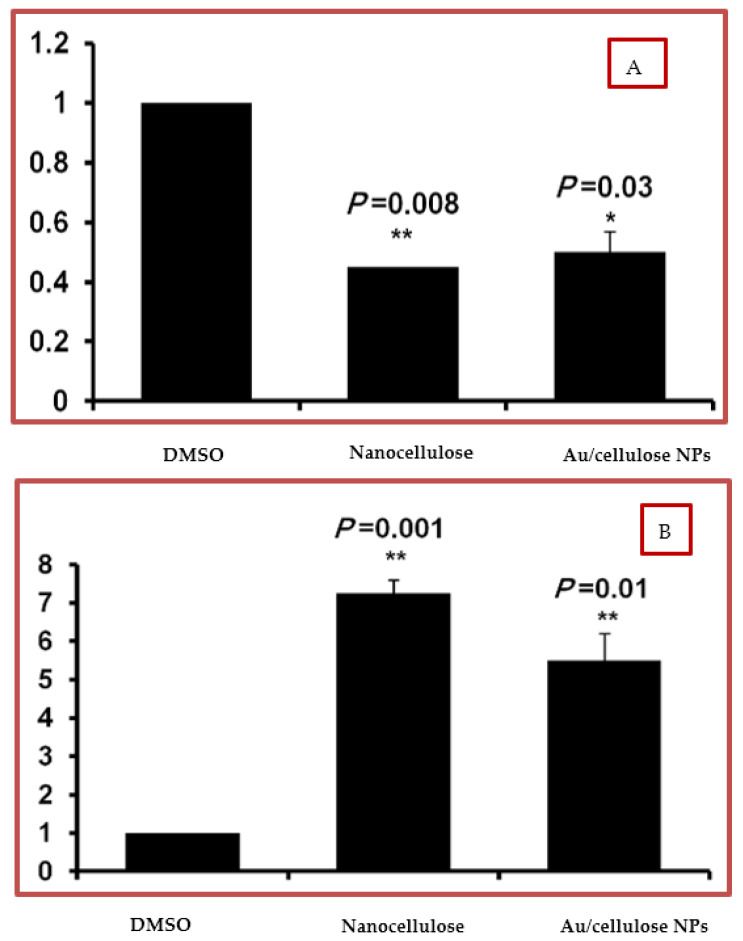
(**A**) Steady-state mRNA of the Raf-1 gene indicated by fold change in nanocellulose, and Au/cellulose nanocomposites compared with DMSO-treated cells. (**B**) Steady-state mRNA of the p53 gene indicated by fold change in nanocellulose and Au/cellulose nanocomposites in comparison with DMSO-treated cells. Levels of GAPDH-mRNA were used as an internal control. * *p* < 0.05 was considered statistically significant and ** *p* < 0.01 as highly significant.

**Figure 9 polymers-13-03340-f009:**
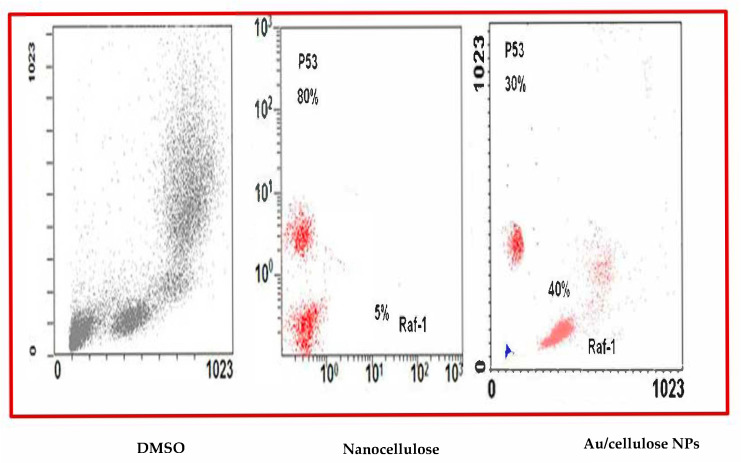
Quantified protein profile of both Raf-1 and pho-P53 in A549 treated by nanocellulose and Au/cellulose nanocomposite indicated by flow cytometry and compared to DMSO-treated cells.

**Table 1 polymers-13-03340-t001:** IC50 values of AuNPs, nanocellulose, and Au/cellulose nanocomposite on normal lung fibroblast HEL299 and lung cancer cells A549.

Treatment	IC50 µg/mL	
	HEL299	A549
AuNPs	95.668 ± 3.59	13.43 ± 0.76
Nanocellulose	2055.9 ± 51.19	86.61 ± 3.21
Au/cellulose nanocomposite	182.75 ± 6.45	4.67 ± 0.17

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
