# Peer review of "Characterization and Anticancer Activity of Biosynthesized Au/Cellulose Nanocomposite from Chlorella vulgaris"

_polymers, 2021, doi:10.3390/polym13193340_

Round 1
Reviewer 1 Report
This paper's aim was to provide information about Au/cellulose nanocomposite with enhanced anticancer activity through the regulation of the MAPK signaling pathway. The manuscript fits within the scope of the journal. The manuscript is interesting and very well organized. The title is clear and it is adequate to the content of the article. The author’s work on discussing achieved results is appreciated.
The revisions are necessary to improve the clarity of the presentation.
- Language style of the paper should be improved
- Please highlight in the introduction the degree of novelty and originality of the work.
- Please use italic for the scientific name.
- L117, 124: please use subscript for formula; see in all text.
- L124 please change ml with mL; see in all text.
- Please explain in detail all methods used for nanoproducts characterization. This form is too brief.
- Please improve the quality of figure 4
- Include in the conclusion potential research directions.
Author Response
Reviewer 1
Comments and Suggestions for Authors
This paper's aim was to provide information about Au/cellulose nanocomposite with enhanced anticancer activity through the regulation of the MAPK signaling pathway. The manuscript fits within the scope of the journal. The manuscript is interesting and very well organized. The title is clear and it is adequate to the content of the article. The author’s work on discussing achieved results is appreciated.The revisions are necessary to improve the clarity of the presentation.
- Language style of the paper should be improved
Thanks for reviewer’s comments
I tried to modify Language style and rephrase sentences as much as possible.
- Please highlight in the introduction the degree of novelty and originality of the work.
Thanks for reviewer’s comments
The degree of novelty and originality of the work has been highlighted in itroduction.
- Please use italic for the scientific name
Thanks for reviewer’s comments
Italic has been used for the scientific name
- L117, 124: please use subscript for formula; see in all text.
Thanks for reviewer’s comments
Subscripts have been used.
- L124 please change ml with mL; see in all text.
Thanks for reviewer’s comments
mL has been used instead of ml
- Please explain in detail all methods used for nanoproducts characterization. This form is too brief.
Thanks for reviewer’s comments
Methods used for nanoproducts characterization have been explained in details.
- Please improve the quality of figure 4
Thanks for reviewer’s comments
The figure quality has been improved.
- Include in the conclusion potential research directions.
Thanks for reviewer’s comments
Potential research directions have been included in the conclusion.

Reviewer 2 Report
The submitted manuscript describes effects of Au nanoparticles nanocellulose and Au/nanocellulose composites on A549 lung cancer cells and HEL299 normal lung fibroblasts. The particles were physicochemically characterized by TEM, UV-Vis, DLS and FTIR. Cellular action was identified by assessment of cytotoxicity and morphology. As potential mode of action mRNA expression of Raf-1 and p53
Comments
- Why were Au nanoparticles at the concentration present in the composites not included in the comparison?
- It should be stated that HEL299 are lung fibroblasts, not normal lung cells because the lung is composed of 40 different cell types
- It is not stated, in which medium the particles were applied to the cells and if they were characterized in this medium or in distilled water.
- The LDH assay determines only cytotoxic action (disruption of plasma membrane integrity) not any other effects that may decrease cell viability, such as decreased proliferation or apoptosis. A conventional viability assay, such as dehydrogenase activity or total protein is needed to find out if disruption of membrane integrity is the only mechanism of action.
- Are changes in Raf-1 and p53 induced also in HEL299?
Minor
There are several spelling errors, e.g. L. 370
Author Response
Reviewer 2
Comments and Suggestions for Authors
The submitted manuscript describes effects of Au nanoparticles nanocellulose and Au/nanocellulose composites on A549 lung cancer cells and HEL299 normal lung fibroblasts. The particles were physicochemically characterized by TEM, UV-Vis, DLS and FTIR. Cellular action was identified by assessment of cytotoxicity and morphology. As potential mode of action mRNA expression of Raf-1 and p53
Comments
- Why were Au nanoparticles at the concentration present in the composites not included in the comparison?
Thanks for reviewer’s comments
Au nanoparticles were included in comparison see figures (5,6 and &7) and Table 1
- It should be stated that HEL299 are lung fibroblasts, not normal lung cells because the lung is composed of 40 different cell types
Thanks for reviewer’s comments
We stated that HEL299 are lung fibroblasts
- It is not stated, in which medium the particles were applied to the cells and if they were characterized in this medium or in distilled water.
Thanks for reviewer’s comments
Cytotoxicity of AuNPs, nanocellulose, and an Au/cellulose nanocomposite on lung cancer A549 cells and on normal lung fibroblasts HEL299 was determined using LDH kit. 40 µl of (0.39, 1.56, 6.3, 25, and 100) µg /ml of AuNPs, nanocellulose and an Au/cellulose nanocomposite in RPMI media were incubated for 1 hour with 40 µl LDH buffer and 20 l µLDH substrate in a 96-well plate
- The LDH assay determines only cytotoxic action (disruption of plasma membrane integrity) not any other effects that may decrease cell viability, such as decreased proliferation or apoptosis. A conventional viability assay, such as dehydrogenase activity or total protein is needed to find out if disruption of membrane integrity is the only mechanism of action.
We completely agree with the reviewer regarding the LDH production point; however, the LDH production somehow indicates the mitochondrial stress that might be cured upon treatment in cancer cells. At the same time, we also try to get a suitable fund to do some experiments related to cell viability using MTT assay or Annex V but unfortunately, we cannot get this fund till now. So, we plan to do some experiments in vivo using a xenograft mouse model and focus on the mechanism of how this compound makes a cytotoxic effect in cancer cells and may be potentially safe to the normal cells.
- Are changes in Raf-1 and p53 induced also in HEL299?
We haven’t ‘performed quantitative real-time polymerase chain reaction analysis and flowcytometry, in HEL299 due to its high cost. we also plan to do this experiment in the Mice xenograft model. But we appreciate your support in getting this work published to support our status in the project that we applied for to have an excellent chance to prepare our project
Minor
There are several spelling errors, e.g. L. 370
Thanks for reviewer’s comments
Errors have been corrected in revised manuscript.

Reviewer 3 Report
The manuscript “Characterization and anti-cancer activity of biosynthesized Au/cellulose nanocomposite from Chlorella vulgaris” deals with the production of nanocellulose, nanogold, and Au/cellulose nanocomposite from microgreen alga Chlorella vulgaris. The work is innovative and ambitious. However, some revisions are required, as follows:
Check that all acronyms are defined in the text.
The quality of all figures has to be improved.
Use subscripts in H2SO4, AuCl4, etc…
Use mL instead of ml.
The production method used for nanoparticles/nanocomposites should be compared with other methods to highlight the advantages/disadvantages. For instance, see this work: Baldino et al., Production, characterization and testing of antibacterial PVA membranes loaded with HA-Ag3PO4 nanoparticles, produced by SC-CO2 phase inversion, Journal of Chemical Technology and Biotechnology, 2019, 94(1), pp. 98–108.
Conclusions are a summary of the work. Rewrite in a more critical way.
English has to be revised.
Author Response
Reviewer 3
Comments and Suggestions for Authors
The manuscript “Characterization and anti-cancer activity of biosynthesized Au/cellulose nanocomposite from Chlorella vulgaris” deals with the production of nanocellulose, nanogold, and Au/cellulose nanocomposite from microgreen alga Chlorella vulgaris. The work is innovative and ambitious. However, some revisions are required, as follows:
- Check that all acronyms are defined in the text
Thanks for reviewer’s comments
all acronyms are defined in the text
- The quality of all figures has to be improved.
Thanks for reviewer’s comments
The quality of all figures has to be improved
- Use subscripts in H2SO4, AuCl4, etc
Subscripts have been used …
- Use mL instead of ml.
Thanks for reviewer’s comments
mL has been used instead of ml
- The production method used for nanoparticles/nanocomposites should be compared with other methods to highlight the advantages/disadvantages. For instance, see this work: Baldino et al., Production, characterization and testing of antibacterial PVA membranes loaded with HA-Ag3PO4 nanoparticles, produced by SC-CO2 phase inversion, Journal of Chemical Technology and Biotechnology, 2019, 94(1), pp. 98–108.
Thanks for reviewer’s comments
Advantage of using cellulose was mentioned in the revised manuscript.
A nanocomposite is produced when nanocellulose and gold nanoparticles are mixed. Cellulose has been utilised as a soft matrix to hold inorganic fillers like metal nanoparticles (Au, Ag, and Cu) in order to create composites that combine the intrinsic functions of the fillers with the biointerfaces provided by cellulose fibres . Cellulose-fiber-reinforced polymer composites have received much attention because of their low density, nonabrasive, combustible, nontoxic, low cost, and biodegradable properties.
- Conclusions are a summary of the work. Rewrite in a more critical way.
Thanks for reviewer’s comments
Conclusion has been rewriten in a more critical way.
- English has to be revised.
Thanks for reviewer’s comments
English has been revised.

Round 2
Reviewer 3 Report
The authors answered all Reviewer's comments. Howerver, in the new paragraph they added (L53-58) according to Reviewer suggestion, ref https://doi.org/10.1002/jctb.5749 has been forgotten.